# Interaction between *KLOTHO*-VS Heterozygosity and *APOE* ε4 Allele Predicts Rate of Cognitive Decline in Late-Onset Alzheimer’s Disease

**DOI:** 10.3390/genes14040917

**Published:** 2023-04-15

**Authors:** Xi Richard Chen, Yongzhao Shao, Martin J. Sadowski

**Affiliations:** 1School of Medicine & Dentistry, University of Rochester, Rochester, NY 14642, USA; 2Department of Population Health, NYU Grossman School of Medicine, New York, NY 10016, USA; 3Department of Environmental Medicine, NYU Grossman School of Medicine, New York, NY 10016, USA; 4Department of Neurology, NYU Grossman School of Medicine, New York, NY 10016, USA; 5Department of Psychiatry, NYU Grossman School of Medicine, New York, NY 10016, USA; 6Department of Biochemistry and Molecular Pharmacology, NYU Grossman School of Medicine, New York, NY 10016, USA

**Keywords:** Alzheimer’s disease, *KLOTHO*, *APOE*, cognitive decline, aging

## Abstract

*KLOTHO*-VS heterozygosity (*KL*-VS^het+^) promotes longevity and protects against cognitive decline in aging. To determine whether *KL*-VS^het+^ mitigates Alzheimer’s disease (AD) progression, we used longitudinal linear-mixed models to compare the rate of change in multiple cognitive measures in AD patients stratified by *APOE ε4* carrier status. We aggregated data on 665 participants (208 *KL*-VS^het−^/*ε4−*, 307 *KL*-VS^het−^/*ε4+*, 66 *KL*-VS^het+^/*ε4−*, and 84 *KL*-VS^het+^/*ε4+*) from two prospective cohorts, the National Alzheimer’s Coordinating Center and the Alzheimer’s Disease Neuroimaging Initiative. All participants were initially diagnosed with mild cognitive impairment, later developed AD dementia during the study, and had at least three subsequent visits. *KL*-VS^het+^ conferred slower cognitive decline in *ε4* non-carriers (+0.287 MMSE points/year, *p* = 0.001; −0.104 CDR-SB points/year, *p* = 0.026; −0.042 ADCOMS points/year, *p* < 0.001) but not in *ε4* carriers who generally had faster rates of decline than non-carriers. Stratified analyses showed that the protective effect of *KL*-VS^het+^ was particularly prominent in male participants, those who were older than the median baseline age of 76 years, or those who had an education level of at least 16 years. For the first time, our study provides evidence that *KL*-VS^het+^ status has a protective effect on AD progression and interacts with the *ε4* allele.

## 1. Introduction

Klotho is a transmembrane protein that is involved in promoting longevity in mammals. The *KLOTHO* (*KL*) gene, which is found in chromosome 13q12, has major and minor alleles [1]. The minor allele is a haplotype of two single nucleotide polymorphisms (SNPs), with the IDs rs9536314 and rs9527025, which remain in perfect linkage disequilibrium and result in F352V and C370S amino acid substitutions, respectively. Heterozygosity for the minor allele (*KL*-VS^het+^) occurs in 18.7% to 25.7% of newborns across ethnically diverse populations and has been associated with extended lifespan [2], while *KL*-VS homozygosity, present in 3% of the population, has been paradoxically linked to survival disadvantage [3]. The effects of klotho on lifespan depends on the concentration of its secretory fragment, known as soluble klotho, which is produced by proteolytic cleavage of the transmembrane holoprotein precursor. The level of secreted klotho is increased in *KL*-VS heterozygotes and conversely reduced in *KL*-VS homozygotes compared to major allele homozygotes [2,4,5]. Transgenic overexpression of klotho in mice has been shown to extend life span [4], while conversely, disruption of the *KL* gene has been associated with an accelerated aging phenotype, which includes atherosclerosis, emphysema, osteoporosis, infertility, and behavioral impairment [5]. The longevity-promoting effects of soluble klotho have also been linked to its antioxidant and anti-inflammatory functions [6,7]. 

Due to its high expression in the mammalian central nervous system (CNS), klotho has been the subject of numerous studies investigating its potential role in cognitive aging. Based on their study of three independent populations of cognitively normal individuals, Dubal and colleagues linked *KL*-VS^het+^ with better cognitive performance in aging individuals [4]. Transgenic overexpression of klotho in mice also improved their behavioral testing performance, and this effect was associated with enhancement of the long-term potentiation (LTP) and upregulation of the subunit GluN2B of the N-methyl-D-aspartate receptor (NMDAR) in the hippocampus [4]. 

Alzheimer’s disease (AD) is a highly prevalent neurodegenerative disease, and its primary risk factors include advanced age, the *APOE ε4* allele [8], and being female [9,10]. AD is characterized by a progressive decline in cognitive function, which eventually results in patients becoming non-ambulatory, non-verbal, and unable to express emotions [11]. The onset of cognitive deficits in AD is insidious. Many individuals are initially diagnosed with mild cognitive impairment (MCI) and progress to dementia within a few years, when they receive a formal AD diagnosis [12]. AD pathogenesis is set off by the deposition of β-amyloid (Aβ) in the brain, which gives rise to the accumulation of hyperphosphorylated tau protein inside neurons [13]. Hyperphosphorylated tau assembles into neurofibrillary tangles (NFTs), and NFT-bearing neurons are then targeted by microglia expressing a specific neurodegenerative phenotype. By creating a local microenvironment with high levels of proinflammatory cytokines and oxidoreductive stress, these microglia further NFT formation and precipitate neuronal death [14,15,16].

The rate of cognitive decline in AD patients is influenced by both the rate of neurodegeneration and the individual’s cognitive reserve. Cognitive reserve is a complex concept that is closely related to the strength of synaptic plasticity response, which acts as a resilience mechanism to counteract the effects of neurodegeneration and protect individuals from exhibiting symptoms of AD [17,18]. Since klotho is strongly implicated in synaptic plasticity response and *KL*-VS^het+^ status has been found to protect against cognitive aging [4], we hypothesize that *KL*-VS^het+^ may slow the progression of cognitive symptoms in AD. Recent studies suggest that *KL*-VS^het+^ protects against NFT pathology [19,20], indicating that it is important to further investigate the effects of *KL*-VS^het+^ on the clinical progression of AD. To this end, we compared the rate of cognitive decline in AD patients enrolled in two prospective AD cohorts, the National Alzheimer’s Coordinating Center (NACC) [21] and the Alzheimer’s Disease Neuroimaging Initiative (ADNI) [22], who were stratified by *KL*-VS^het+^ and *APOE ε4* carrier status. We selected the *ε4* allele as a co-variate and examined its interactions with *KL*-VS^het+^ on AD progression for our analysis, not only because it is the strongest genetic risk factor for sporadic AD, but also because it is known to increase the rate of dementia progression in an allele dose-dependent manner in patients with established disease [23,24,25]. We utilized linear-mixed models (LMM) to analyze longitudinal cognitive data during the MCI and dementia stages of AD. Our analysis revealed a protective effect of *KL*-VS^het+^ on AD progression, and we also observed an interaction between *KL*-VS^het+^ and the *ε4* allele, which is a novel gene–gene interaction specific to the disease.

## 2. Materials and Methods

### 2.1. Participant Selection

Demographic information, longitudinal cognitive measures, clinical diagnosis (MCI vs. AD dementia), and genetic information were retrieved from the NACC and ADNI databases. Both the NACC and ADNI are independent, multicenter, prospective studies that investigate the natural progression of AD and also provide neuropathological verification of clinical diagnoses. The studies were approved by the Institutional Review Boards of all institutions that contributed participants. Prior to the studies, participants or their caregivers provided informed written consent. Participant information from the NACC was retrieved via the Uniform Data Set (UDS), which includes longitudinal phenotype data, from the June 2022 data freeze. The UDS is a data repository that contains annual clinical evaluations of participants who were recruited by NIA Alzheimer’s Disease Researcher Centers (ADRC) programs starting in 2005 and continuing to the present day. Each program has its own protocol for referrals and recruitment, and a more detailed summary of NACC data compilation is available at https://naccdata.org/requesting-data/data-request-process#naccHandbook (accessed on 20 March 2023). From the ADNI, data was extracted on 23 January 2022 and included data from its four consecutive studies: ADNI-1 (2004–2009), ADNI-GO (2009–2011), ADNI-2 (2011–2016), and ADNI-3 (2016–present). These are successive studies that emphasize participant rollover with additional recruitment goals for each study. Complete ADNI criteria and data collection information are available at http://adni.loni.usc.edu/data-samples/clinical-data/ (accessed on 23 January 2022) and https://adni.loni.usc.edu/methods/documents/ (accessed on 23 January 2022). 

Participants selected for the analysis met our previously published criteria, ensuring they presented a cognitive decline that is consistent with clinical AD progression and had a sufficient number of data points to allow for longitudinal modeling of the rate of cognitive decline [23]. Specific criteria included: (1) entering the study with an MCI diagnosis and transitioning to an AD dementia during the study, (2) having at least three follow-up visits after receiving an AD dementia diagnosis, (3) not reverting the diagnosis from AD dementia to MCI or normal, and also (4) having both known *KL* and *APOE* genotype statuses. In the ADNI, subjects were diagnosed with MCI if there was a memory complaint by either the participant or the study partner, memory loss measured by education-adjusted scores on the Weschler Memory Scale (Logical Memory II subscale), a score between 24 and 30 (inclusive) on the MMSE, a score of 0.5 on the Global CDR, and preserved general cognition and functional performance, as determined by the site physician. A diagnosis of AD was recorded if participants scored between 20 and 26 (inclusive) on the MMSE, either 0.5 or 1.0 on the Global CDR, and met the NINCDS/ADRDA criteria for probable AD. In the NACC, clinicians were instructed to assess cognition with neuropsychological testing of their choosing, only providing commonly used cut-off points such as a score of 0.5 on the CDR representing MCI and a score of 1.0 or above representing AD. Both the ADNI and NACC also considered other neurological conditions that may contribute to or directly cause MCI or dementia. Participants with these neurological co-morbidities were excluded from analysis. 

Non-white and Latino individuals were excluded from the analysis since they represented less than 5% of the initially identified participants. This was done to increase homogeneity in the analyzed cohort and reduce the potential confounding effect of population stratification, which is consistent with similar studies [26,27]. After processing genetic data as described below, 665 participants from the combined ADNI and NACC cohorts were selected for final analysis. A total of 497 participants (74.7%) were homozygous for the major *KL* allele, 18 (2.7%) were homozygous for the minor *KL* allele, and 150 (22.6%) were heterozygous for the minor *KL* allele. *APOE* genotype frequency was as follows: 253 *ε3*/*ε3* (38.0%), 285 *ε3*/*ε4* (42.9%), 22 *ε3*/*ε2* (3.3%), 90 *ε4*/*ε4* (13.5%), and 15 *ε4*/*ε2* (2.3%). Combining the groups, we found that 208 were *KL*-VS^het^/*ε4−* (31.3%), 307 were *KL*-VS^het−^/*ε4*+ (46.2%), 66 were *KL*-VS^het+^/*ε4−* (9.9%), and 84 were *KL*-VS^het+^/*ε4+* (12.6%) (Table 1).

### 2.2. Genetic Data Quality Control and Processing

Genomic datasets were analyzed using Plink 1.9 software developed by the Purcell Lab in Boston, MA, and updated by Christopher Chang with support from the NIH-NIDDK’s Laboratory of Biological Modeling in Bethesda, MD [28]. Genetic data for NACC participants were provided by The National Institute on Aging Genetics of Alzheimer’s Disease Data Storage Site (NIAGADS). Genetic data for ADNI participants were retrieved from the Laboratory of Neuro Imaging (LONI) website hosted by the University of Southern California. Both datasets were subject to basic quality control on all data pre-imputation, filtering out variants with a minor allele frequency (MAF) < 1%, and excluding variants with an imputation R^2^ < 0.04. Subjects were excluded if they had autosome missingness ≥5%, discrepancies between recorded sex and sex based on expected X chromosome heterozygosity/homozygosity rates (>0.8 X chromosome homozygosity estimate for males, <0.2 for females), and heterozygosity rates above or below three standard deviations of the mean rate. SNPs were excluded for a call rate ≤ 95%, MAF ≤ 1%, and deviation from the Hardy–Weinberg distribution with a significance threshold of *p* < 5 × 10^−5^. Identity by descent was calculated with a threshold of 0.2, allowing us to detect and exclude subjects who are second-degree relatives. Only one subject per relatedness group was randomly included for analysis. Multidimensional scaling was then applied to correct for population stratification, eliminating outliers with 10 main components used as covariates in the association tests. *KL*-VS status was derived from the rs9536314 SNP ID and *APOE* status from subject demographics.

### 2.3. Cognitive Measures

In both the ADNI and NACC datasets, participants underwent an initial evaluation at baseline and were subsequently assessed annually. However, it should be noted that the ADNI dataset includes an additional evaluation at 6 months after the baseline visit. Both studies utilized the Mini-Mental State Examination (MMSE) (ranges from 0 to 30, decreased score indicating worse cognition) and the Clinical Dementia Rating Scale Sum of Boxes (CDR-SB) (ranges from 0 to 18, increased score indicating worse cognition). The MMSE assesses cognitive function across several domains including orientation to time and place, registration, attention and calculation, short-term recall, visuospatial functions, and language skills to screen for cognitive impairment and to approximate its depth. The test is performed by a trained healthcare professional who asks a battery of questions and provides a series of tasks that take approximately 15 min to complete. The CDR serves a similar purpose; however, it evaluates cognition across six domains (memory, orientation, judgement and problem solving, community affairs, home and hobbies, and personal care) based on a semi-structured interview that includes responses from the participant and the participant’s caregiver. CDR-SB is then computed by summing the domain box scores. Participants in the ADNI additionally received the Alzheimer’s Disease Assessment Scale-Cognitive Subscale (ADAS-Cog, also known as ADAS-11) (ranges from 0 to 70, increased score indicating worse cognition), which evaluates cognition through a series of 11 tasks and questions including word recall, orientation, and language comprehension. The ADAS-11 also allows us to calculate the AD Composite Score (ADCOMS) in the ADNI cohort alone. ADCOMS is a composite variable (ranges from 0 to 1.97, increased score indicating worse cognition) derived from selected elements of ADAS-11, MMSE, and CDR-SB that have demonstrated improved sensitivity to longitudinal cognitive decline and reduced inter-testing variability compared to individual scales [29]. A total of 665 participants representing the ADNI and the NACC combined had MMSE and CDR-SB measured, while 178 participants representing the ADNI alone had ADAS-11 measured and therefore had ADCOMS computed.

### 2.4. Statistical Analyses

Demographic and clinical data across the *KL*-VS^het+^/*ε4+*, *KL*-VS^het−^ /*ε4+*, *KL*-VS^het+^/*ε4−*, and *KL*-VS^het−^/*ε4−* groups were analyzed using one-way analysis of variance (ANOVA) followed by Least Significant Difference (LSD) test as a post hoc analysis (Table 1). The frequency of each *APOE* was tabulated in Table 2 and compared between *KL*-VS^het+^ and *KL*-VS^het−^ groups for ε4 carriers and non-carriers using Fisher’s exact tests. This analysis was done to determine whether the incidence of *ε2* or *ε4* alleles could account for observed protective effects of *KL*-VS^het+^ in *ε4* non-carriers or the lack of protective effects in *ε4* carriers, respectively (Table 2). Longitudinal LMM analysis was used to examine how *KL*-VS^het+^ and *ε4* carrier status affect cognitive measures over time. LMM analysis reduces non-random attrition bias by clustering each subject’s repeated visits and modeling random intercepts. This approach allows for the comparison of subjects with different numbers of evaluations by predicting dependent variables based on fixed effects, including intercepts that vary between groups. Time from baseline visit, sex, age at baseline, and years of education were used as fixed effects. For each LMM analysis, we computed the *p* value and the regression coefficient (β) ± standard error (SE). 

A negative main effect of time suggests that there was significant cognitive decline across the entire study cohort, while a main effect of *KL*-VS^het+^ and/or *ε4* carrier status indicates a baseline difference in cognitive performance across the groups. An interaction between time and *KL*-VS^het+^ and/or *ε4* carrier status means that there is a difference in the predicted rates of cognitive decline between sub-groups defined by their *KL*-VS and *APOE* genotypes, with the β value representing the estimated direction and amount of change. We also stratified LMM analyses in *APOE ε4* allele non-carriers by the median baseline age (≤76 years vs. >76 years), sex, and years of education (<16 years vs. ≥16 years) for the combined ADNI and NACC cohort. Annual rates of change for all cognitive measures stratified by *KL*-VS^het+^ and *ε4* carrier status were determined by a multiple linear regression model. Parameter estimates from LMM analysis were used as the dependent regression variables while time was used as the independent variable. 

Unlike most prior studies where the exact age of initial AD diagnosis is usually censored or interval censored, the present dataset has the advantage of precisely identifying the age at which participants transitioned from MCI to AD. For all analyses, longitudinal data from individual participants were aligned by assigning a time variable value of zero to the first visit the diagnosis of AD dementia was made. Consequently, all visits under an MCI diagnosis were given negative values and those under an AD diagnosis given positive values. This approach allows us to precisely anchor any modifying genetic effects to disease onset and eliminate the variance in age at baseline and time between baseline visit and actual AD diagnosis. All statistical analyses were performed using IBM^®^ SPSS^®^ Statistics 25 (IBM Corp., Armonk, NY, USA).

## 3. Results

### 3.1. Descriptive Statistics

Participants in our study were 53.4% male, with an average baseline age of 75.5 years ± 8.1 years (mean ± standard deviation), an average transition age of 77.7 years ± 8.2 years, and 16.2 years ± 6.3 years of education (Table 1). One-way ANOVA analysis indicated a significant difference in age at baseline (*F* = 14.263, *p* = 0.000, df = 3, mean square = 872.252) and age at transition from MCI to AD (*F* = 13.728, *p* = 0.000, df = 3, mean square = 869.222) across groups defined by *KL-*VS^het+^ and *ε4* carrier status. LSD post hoc tests found that the baseline age of participants who were either *KL-*VS^het−^/*ε4−* (77.9 years ± 8.8 years) or *KL-*VS^het+^/*ε4−* (77.7 years ± 8.7 years) was significantly greater than the baseline age of participants who were *KL-*VS^het−^/*ε4+* (74.0 years ± 7.1 years) or *KL-*VS^het+^/*ε4+* (73.2 years ± 7.2 years) (*p* < 0.05). Similarly, the age at transition for participants who were either *KL-*VS^het−^/*ε4−* (80.1 years ± 8.9 years) or *KL-*VS^het+^/*ε4−* (80.3 years ± 9.3 years) was significantly greater than the baseline age of participants who were *KL-*VS^het−^/*ε4+* (76.1 years ± 7.1 years) or *KL-*VS^het+^/*ε4+* (75.3 years ± 7.4 years). No effect of *KL-*VS^het+^ status alone was observed on baseline age (*F* = 0.244, *p* = 0.621, mean square = 15.849) or transition age (*F* = 0.498, *p* = 0.480, mean square = 33.402). We found no differences in the prevalence of particular *APOE* genotypes both between *KL*-VS^het+^/*ε4+* and *KL*-VS^het−^/*ε4+* groups and between *KL*-VS^het+^/*ε4−* and *KL*-VS^het−^/*ε4−* groups (Table 2). There were also no significant differences in sex composition or years of education across the four groups defined via cross-classification by *KL-*VS^het+^ and *ε4* carrier status (Table 1). 

### 3.2. KL-VS Heterozygosity Has a Protective Effect on the Rate of Cognitive Decline in ε4 Non-Carriers

The analysis using longitudinal LMM was conducted on a cohort consisting of both male and female participants, with roughly equal numbers of each sex. This initial cohort was divided based on *KL-*VS^het+^ and *ε4* carrier status, as detailed in Table 1. The analysis revealed a main effect of time on each cognitive measure (*p* < 0.001) (Table 3). There were no significant differences across groups for either MMSE or CDR-SB baseline values. In the ADNI participants, there was a significant difference in baseline ADCOMS score where *KL-*VS^het−^ had a higher score than *KL-*VS^het+^ in *ε4* non-carriers (β = 0.137, *p* = 0.023) and *KL-*VS^het−^ in *ε4* carriers had a higher score than *KL-*VS^het+^ in *ε4* non-carriers (β = 0.145, *p* = 0.012), where the β values represent the estimated difference in baseline scores. 

In all measures, *ε4* carriers showed a faster rate of cognitive decline compared to *ε4* non-carriers. *KL*-VS^het+^ status slowed the rate of progression compared to *KL*-VS^het−^ but only in *ε4* non-carriers. In MMSE, we found that *ε4* carriers had greater rates of decline than *ε4* non-carriers in both *KL*-VS^het+^ (*ε4*+ at −1.358 MMSE/year, *ε4−* at −0.735 MMSE/year, *p* = 0.000) and *KL*-VS^het−^ (*ε4*+ at -1.423 MMSE/year, *ε4−* at -1.067 MMSE/year, *p* = 0.000) individuals (Table 4). Importantly, within *ε4* non-carriers, *KL*-VS^het+^ participants had a slower rate of decline than *KL*-VS^het−^ participants (β =+0.287 point/year, *p* = 0.001). This remains significant even after Bonferroni correction for multiple testing. This effect was not seen in *ε4* carriers (β = −0.036, *p* = 0.659). Considering CDR-SB, *ε4* carriers also declined faster than *ε4* non-carriers in both *KL*-VS^het+^ (*ε4*+ at +1.312 CDR-SB/year, *ε4−* at +0.903 CDR-SB/year, *p* = 0.000) and *KL*-VS^het−^ (*ε4*+ at +1.216 CDR-SB/year, *ε4−* at +1.056 CDR-SB/year, *p* = 0.000) groups. Again, *KL*-VS^het+^ status conferred a slower rate of decline within *ε4* non-carriers (β = −0.104 point/year, *p* = 0.026) but not within *ε4* carriers (β = −0.041 point/year, *p* = 0.362). Similarly, in the ADNI cohort alone, the rate of change in ADCOMS was greater in *ε4* carriers within the *KL*-VS^het+^ (*ε4*+ at +0.122 ADCOMS/year, *ε4−* at +0.047 ADCOMS/year, *p* = 0.000) and *KL*-VS^het−^ (*ε4*+ at +0.112 ADCOMS/year, *ε4−* at +0.082 ADCOMS/year, *p* = 0.000) participants. Those with *KL*-VS^het+^ status showed significantly slower rates of progression within *ε4* non-carriers (β = −0.042 point/year, *p* = 0.000) but again not within *ε4* carriers (β = −0.009 point/year, *p* = 0.243). These interactions between *KL*-VS^het+^ and *APOE ε4* allele are illustrated for the entire mixed-sex cohort in three linear regression plots divided by cognitive measure in Figure 1A–C. Fisher’s exact tests showed no significant differences in the frequency of APOE genotypes between *KL*-VS^het+^ and *KL*-VS^het−^ subjects in either *ε4* carriers (*p* = 0.224) or *ε4* non-carriers (*p* = 0.616). This analysis suggests that the differences in the incidence of the *ε4* or *ε2* alleles do not account for the observed protective effects of *KL*-VS^het+^ in *ε4* non-carriers and the lack of protective effects in *ε4* carriers (Table 2).

### 3.3. The Protective Effect of KL-VS Heterozygosity Is Observed in Males, in Individuals with an Older Age of Cognitive Decline Onset, and in Those with 16 or More Years of Education

Longitudinal LMM analyses were repeated to assess *KL*-VS^het+^ effect in *ε4* non-carriers, who were stratified by sex, age of the baseline visit, and years of education (Table 5). To conduct these analyses, we selected VS^het+^/*ε4−* individuals based on the demographic of interest and compared them to the entire VS^het−^/*ε4−* cohort, which was used as the reference group. Stratified analyses were performed for MMSE and CDR-SB but not for ADCOMS because of limited number of participants with longitudinal ADCOMS data. Male participants with *KL*-VS^het+^ status showed a significantly slower rate of decline compared to the average decline rate in the *KL*-VS^het−^/*ε4−* cohort both in MMSE (β = +0.368 point/year, *p* = 0.001) and CDR-SB (β = −0.196 point/year, *p* = 0.001). There was no significant effect in *KL*-VS^het+^ females on either MMSE (β = +0.039 point/year, *p* = 0.795) or CDR-SB (β = +0.015 point/year, *p* = 0.834). These interactions between *KL*-VS^het+^ and *APOE ε4* allele for MMSE and CDR-SB are illustrated separately for female and male participants using linear regression plots in Figure 2A–D.

Similarly, within *ε4* non-carriers, the *KL*-VS^het+^ status was protective in the participants who were older than 76 years at the baseline visit, significantly slowing progression on MMSE (β = +0.520 point/year, *p* < 0.000) and CDR-SB (β = −0.285 point/year, *p* < 0.000) but had no significant effect in participants who were 76 years or younger at baseline (MMSE β = −0.029 point/year, *p* = 0.867 and CDR-SB β = +0.047 point/year, *p* = 0.621). *KL*-VS^het+^ status also was significantly protective in participants with 16 years of education or higher (MMSE β = +0.287 point/year, *p* = 0.001 and CDR-SB β = −0.104 point/year, *p* = 0.026) but not in participants with less than 16 years of education (MMSE β = −0.025 point/year, *p* = 0.885 and CDR-SB β = +0.136 point/year, *p* = 0.117). We used the median value of 16 years for years of education, but it should be noted that exactly 202 participants had this same level of education. It is also worth noting that the NACC and ADNI datasets generally include highly educated individuals.

## 4. Discussion

Previous studies have shown that *KL*-VS^het+^ status is associated with extended lifespan and protection against age-related cognitive decline [2,3,4]. Our study demonstrates, for the first time, that *KL*-VS^het+^ status also slows down the progression of cognitive decline related to AD, and this effect is dependent on the absence of the APOE *ε4* allele. The LMM analysis, utilizing the advantage of longitudinal measures, revealed a strong protective effect of *KL*-VS^het+^ on MMSE and CDR-SB scales, even with a sample size as small as 274 APOE *ε4* non-carriers. This effect was confirmed on the ADCOMS scale with a smaller sample of 64 *ε4* non-carriers. However, the protective effect was not observed in *ε4* carriers, indicating a previously under-appreciated gene–gene interaction between these prominent genetic factors in aging. As previously observed, AD progression was faster in *ε4* carriers compared to non-carriers, regardless of their *KL*-VS^het+^ status [23].

Klotho and apolipoprotein (apo) E are involved in multiple biological processes both outside and inside the CNS. The apoE4 isoform is uniquely modified by the presence of arginine in positions 112 and 158, resulting in an intramolecular domain interaction absent in other apoE isoforms [30,31,32]. There are several biological mechanisms associated with *KL*-VS^het+^ status that could improve clinical outcomes in AD but the presence of the *ε4* allele is known to adversely affect these mechanisms. One potential mechanism concern opposing effects of *KL*-VS^het+^ and the *ε4* allele on cognitive reserve. Several studies have shown that older, cognitively normal individuals with *KL*-VS^het+^ perform better on a number of cognitive tasks compared to individuals with *KL*-VS^het−^ [4,20,27], while the opposite effect is true of the *ε4* allele, where carriers perform worse than non-carriers [33,34]. The cognitive benefits linked to elevated klotho levels in *KL*-VS^het+^ subjects have been examined using transgenic rodent models. Dubal and colleagues demonstrated that transgenic overexpression of klotho in mice enhances behavioral testing performance through augmentation of NMDAR-related effects, including upregulated FOS expression after learning and memory tasks, amplified LTP response in the hippocampus, and upregulated expression of the NMDAR subunit GluN2B both in the hippocampus and cortex [4]. Conversely, research on mice with disrupted *KL* genes has shown significant behavioral deficits associated with synaptic and cytoskeletal dysfunction [35,36,37]. Examples of *ε4*–related mechanisms that are detrimental to cognitive performance during aging and AD include hyperexcitability of hippocampal networks [38], age-related loss of hippocampal interneurons [39,40], and dysfunctional endosomal trafficking, resulting in reduced surface expression of NMDAR subunits [41,42]. Interestingly, Dubal and colleagues have noted that the protective effects of *KL*-VS^het+^ on cognition during normal aging are independent of *ε4* allele status [4]. This observation suggests that in the absence of AD pathology, *KL*-VS^het+^-related benefits can counteract various otherwise detrimental *ε4*-related effects on synaptic plasticity during aging.

The second group of biological effects antagonistically driven by *KL*-VS^het+^ status and the *ε4* allele includes oxidative stress and neuroinflammation. The klotho protein has an anti-aging effect, particularly in the hippocampus, by protecting against the oxidation of DNA and membrane lipids [6,7,43]. Mice with a disrupted *KL* gene exhibit a pathological phenotype that includes behavioral deficit, as demonstrated by worse performance on the novel object recognition test, a hippocampal-dependent task. This deficit can be dramatically improved by treatment with the antioxidant α-tocopherol [37]. Klotho also has a direct anti-inflammatory property by counteracting the effects of TNFα, which is a potent proinflammatory cytokine [44]. Upregulation of the TNFα level is associated with normal aging, while in AD pathogenesis, TNFα is a key driver of microglia and astrocyte inflammatory activation [15,45]. In AD, glia-driven inflammation is associated with substantial levels of oxidative injury. Both neuroinflammation and oxidative stress are leading mechanisms promoting NFT formation and neuronal death [25]. Therefore, the mitigating effect of *KL*-VS^het+^ on cognitive decline we observed in *ε4* non-carriers can be partly attributed to klotho’s anti-oxidant and anti-inflammatory effects. In contrast, apoE4 has been recognized as a catalyst of neuroinflammation, with its high expression level being a hallmark of microglia that assume a neurodegenerative phenotype [14,15,25]. Therefore, it is plausible that in *KL*-VS heterozygotes, the strong proinflammatory effects of apoE4 counteract anti-inflammatory and anti-oxidant effects endowed by increased levels of secreted klotho.

Thirdly, *KL*-VS^het+^ and the *APOE ε4* allele show opposing effects on the progression of NFT pathology. Recent work by Driscoll and colleagues found lower levels of total and phospho-tau in the cerebrospinal fluid (CSF) of older, cognitively normal *KL*-VS^het+^ individuals [20]. Meanwhile, Neitzel and colleagues demonstrated that *KL*-VS^het+^ individuals who develop symptomatic AD have lower NFT burden, controlled for Aβ load [19]. Their study was based on the assumption that initial Aβ deposition gives rise to NFT pathology and becomes its effective driver, at least in the early stage of the process [46]. In contrast to *KL*-VS^het+^, the *ε4* allele has been associated with a more aggressive course of NFT pathology, evidenced in neuroimaging studies in AD patients [47,48,49] and transgenic animal model experiments [25,50]. Interestingly, two independent studies have found that *KL*-VS^het+^ protective effects in Aβ-induced NFT pathology benefits *ε4* carriers relatively more than non-carriers, despite the fact that the former have greater NFT loads [19,51]. The mechanisms responsible for the protective effect of *KL*-VS^het+^ against NFT pathology have not yet been fully analyzed in AD transgenic mouse models and remain speculative. However, they may include klotho’s pro-autophagic effect [52], which has been shown to be involved in the clearance of intracellular tau aggregates [53]. As previously discussed, klotho’s anti-oxidant and anti-inflammatory properties naturally mitigate NFT pathology, while the *APOE ε4* allele has been well documented to be involved in several mechanisms that promote NFT pathology. These include impaired autophagy of hyperphosphorylated tau [54], redistribution of hyperphosphorylated tau from axons to cell bodies, which propagates NFT formation within neurons [27], and the previously highlighted faciliatory effect of *APOE ε4* on the transformation of resting microglia to a neurodegenerative phenotype, which is an eminent driver of neuroinflammation and neurodegeneration [14,15,16]. 

Lastly, the potential influence of the APOE genotype on the level of secreted klotho protein should be taken into consideration. Secreted klotho is abundant in the serum, where its main source are the kidneys, and in the CSF, where it is produced by the choroid plexus. To what extent klotho level correlates in these two compartments is unknown. Klotho concentration in the serum decreases with advancing age, and it also inversely varies with renal function parameters such as creatinine and blood urea nitrogen levels [55]. Similarly, its level in the CSF also decreases with advanced age, and in AD patients, CSF klotho levels have been found to be markedly suppressed [56]. Unfortunately, there is a lack of data on whether the severity of AD pathology, which is associated with the APOE genotype, correlates inversely with CSF klotho concentration. Therefore, further studies are necessary to clarify the changes in CSF klotho levels during the course of AD. These studies should stratify participants based on their *APOE ε4* allele status to determine whether the *ε4* allele intensifies the AD-associated reduction in CSF klotho levels. 

Our stratified analyses suggest that AD-protective effects of *KL*-VS^het+^ in *ε4* non-carriers are associated with male sex, later onset of cognitive impairment, and education exceeding 16 years. Several factors could explain the lack of *KL*-VS^het+^ protection in women, including increased vulnerability to NFT pathology, the adverse effect of post-menopausal hormonal deficiency on cognitive reserve, and lower levels of secreted klotho protein. Compared to males, female AD patients tend to accumulate more NFTs [49,57,58] and experience more severe NFT-associated brain atrophy [59]. A recently identified explanation for this sexual dimorphism involves the higher level of ubiquitin-specific peptidase 11 in women as compared to men. Ubiquitin-specific peptidase 11 is an X-linked enzyme responsible for tau deubiquitination, and therefore more ubiquitin removal promotes tau aggregation [60]. The postmenopausal drop in sex hormone levels has been well-established as detrimental to cognitive reserve in aging females, as these hormones are strongly implicated in hippocampal plasticity mechanisms [61]. This has been confirmed by several cross-sectional studies documenting worse performance on a number of cognitive tasks in older females who are at risk of AD compared to age-matched males [62,63,64]. Additionally, female subjects have been found to exhibit lower concentration of secreted klotho in the CSF compared to male subjects [56]. The reasons for sexual dimorphism in CSF klotho level is unclear, but similar effects have been observed in transgenic mice overexpressing klotho, where concentrations of the transgenic protein were lower in females than in male animals. These differences had a measurable impact on rodent lifespan, which was extended by 20% in females compared to 30% in males [65]. 

Our analysis revealed that the protective effect of *KL*-VS^het+^ among *ε4* non-carriers is associated with later onset of cognitive impairment. We used the median baseline visit age, which was 76 years, to divide participants into older and younger cohorts who had MCI but not dementia due to AD. This is not to suggest that the *KL*-VS^het+^ effect is sharply demarcated at 76 years. However, our limited sample size precluded more granular, step-wise, age-dependent stratification. The effect of *KL*-VS^het+^ on older participants is likely related to the more indolent course of AD pathology in comparison to younger individuals, who have a more aggressive disease progression. This is consistent with our previous finding that the *ε4* allele accelerates AD progression in individuals who transition to dementia before the age of 76.1 years, but not in those who transition later than that [23]. Similarly, the *KL*-VS^het+^ protective effect associated with 16 or more years of education can be reasonably explained as an interaction with greater cognitive reserve capacity among higher-educated individuals [66]. 

In addition to the effects of klotho on cognitive reserve, neuroinflammation, and NFT pathology discussed earlier, Belloy and colleagues have provided evidence that *KL*-VS^het+^ status may modulate overall risk of AD [67]. They found that *KL*-VS^het+^ status is associated with reduced AD risk in individuals carrying the *APOE ε4* allele, with the effect being most pronounced between ages of 60 and 80 years. They also showed that *KL*-VS^het+^ status is associated with an attenuated accumulation of Aβ in the brains of cognitively normal *ε4* carriers but not in *ε4* non-carriers, providing a plausible explanation for the mitigating effect of *KL*-VS^het+^ on the disease risk in the individuals carrying the *ε4* allele [26,67]. Although the biological mechanisms underpinning klotho’s effect on Aβ pathology remain hypothetical, they may include the regulatory effect of soluble amyloid precursor protein on klotho protein expression [68] and klotho’s activating effect on autophagy [52], which protects against the accumulation of misfolded proteins in AD [53]. However, the mechanisms underlying the favorable interactions between klotho and the apoE4 isoform on Aβ deposition remain yet to be ascertained.

## 5. Conclusions

Our work reveals that *KL*-VS^het+^ status enhances resilience to AD-related cognitive decline in male patients who do not carry the *APOE ε4* allele. This protective effect is not observed in female *ε4* non-carriers and is nullified by the presence of the *ε4* allele in both male and female subjects. Klotho and apoE have complex, opposing effects on many mechanisms implicated in AD pathogenesis. We propose that the sum of these effects is balanced in *KL*-VS^het+^ individuals carrying the *APOE ε4* allele, resulting in no protective effect. While biomarker studies suggest that *KL*-VS^het+^ status benefits *ε4* allele carriers relatively more than non-carriers with respect to Aβ and NFT accumulation, the negative effects of the *ε4* allele on cognitive reserve, neuroinflammation, and oxidative stress ultimately outweigh any benefits conferred by *KL*-VS^het+^. The lack of a *KL*-VS^het+^ benefit in female subjects who are *ε4* non-carriers suggests that females may be more vulnerable to the mechanisms of neurodegeneration in AD.

## Figures and Tables

**Figure 1 genes-14-00917-f001:**
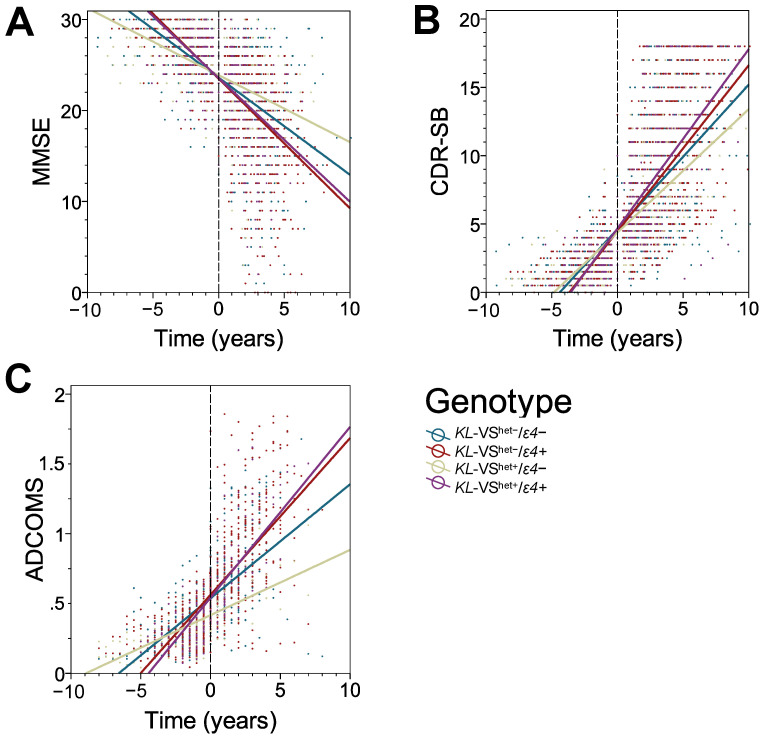
*KL*-VS^het+^ status significantly reduces rate of cognitive decline in AD patients who are *APOE ε4* allele non-carriers but not in those who carry the *ε4* allele. Shown are nonparallel linear regression plots with distinctive slopes reflecting the interaction between *KL*-VS^het+^ status and *APOE ε4* allele carrier status in the complete, sex-mixed cohort on Mini-Mental State Examination (MMSE) (**A**), Clinical Dementia Rating Sum of Boxes (CDR-SB) (**B**), and AD Composite Score (ADCOMS) (**C**). Solid colored lines represent lines of best fit through individual data points in a scatterplot. Negative time variable values represent the number of years before transition from MCI to AD dementia, and positive values represent the number of years after the transition to AD dementia. The 0 values represent the clinical visit a participant for the first time received a diagnosis of AD dementia.

**Figure 2 genes-14-00917-f002:**
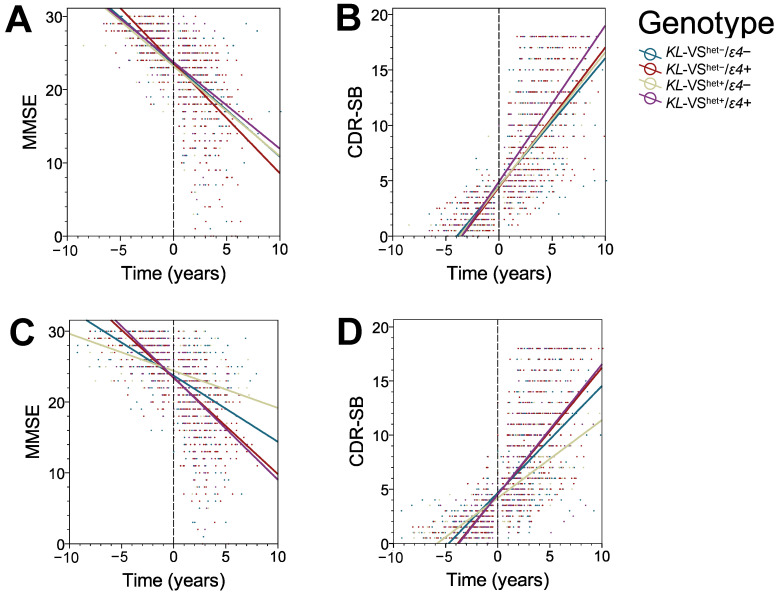
Protective effect of the *KL*-VS^het+^ status is clearly appreciated in male but not in female participants, who are *APOE ε4* allele non-carriers. Shown are nonparallel linear regression plots with distinctive slopes reflecting the interaction between *KL*-VS^het+^ status and *APOE ε4* allele carrier status on Mini-Mental State Examination (MMSE) (**A**,**C**), and Clinical Dementia Rating Sum of Boxes (CDR-SB) (**B**,**D**) separately in female (**A**,**B**) and male (**C**,**D**) participants. Solid colored lines represent lines of best fit through individual data points in a scatterplot. Negative time variable values represent the number of years before transition from MCI to AD dementia, and positive values represent the number of years after the transition to AD dementia. The 0 values represent the clinical visit a participant for the first time received a diagnosis of AD dementia.

**Table 1 genes-14-00917-t001:** Demographic and clinical characteristics in analyzed participants and grouped by *KL-VS^het^* and *APOE ε4* status.

Parameter	All (*n* = 665)	*KL-*VS^het−^/*ε4−* (*n* = 208)	*KL-*VS^het−^/*ε4+* (*n* = 307)	*KL-*VS^het+^/*ε4−* (*n* = 66)	*KL-*VS^het+^/*ε4+* (*n* = 84)
Number of visits	10.7 (4.1)	10.1 (3.8)	11.1 (4.0)	12.1 (5.9)	10.0 (4.3)
Baseline age (years) ***	75.5 (8.1)	77.9 (8.8)	74.0 (7.1)	77.7 (8.7)	73.2 (7.2)
Transition age (years) ***	77.7 (8.2)	80.1 (8.9)	76.1 (7.1)	80.3 (9.3)	75.3 (7.4)
Education (years)	16.2 (6.3)	16.8 (8.7)	15.8 (2.8)	15.6 (2.9)	16.8 (9.4)
% Male	53.4%	55.8%	51.8%	48.5%	57.1%

“Baseline age” refers to the age of participants at their baseline visit, whereas “Transition age” is the age during the visit when they receive the diagnosis of AD dementia. All data are either presented as mean values with standard deviation in parentheses or as percentages where applicable. *** indicates *p* < 0.001 from one-way analysis of variance across *KL-*VS^het−^/*ε4−*, *KL-*VS^het−^/*ε4+*, *KL-*VS^het+^/*ε4−*, and *KL-*VS^het+^/*ε4+* cohorts. Results of pairwise post hoc analysis for parameters with significant one-way analysis of variance are described in Section 3.1.

**Table 2 genes-14-00917-t002:** Total and relative, in parentheses, frequencies of particular *APOE* genotypes in groups defined by *KL-*VS^het^ and *APOE ε4*-carrier status.

	*APOE ε4−* (*p* = 0.794)		*APOE ε4+* (*p* = 0.792)	
*APOE* **Genotype**	*KL*-VS^het−^	*KL*-VS^het+^	*KL*-VS^het−^	*KL*-VS^het+^
*ε2*/*ε2*	0 (0%)	0 (0%)	-	-
*ε2*/*ε3*	16 (7.7%)	6 (9.1%)	-	-
*ε3*/*ε3*	193 (92.3%)	60 (90.9%)	-	-
*ε2*/*ε4*	-	-	12 (3.9%)	3 (3.6%)
*ε3*/*ε4*	-	-	221 (72.2%)	64 (76.2%)
*ε4*/*ε4*	-	-	73 (23.9%)	17 (20.2%)

**Table 3 genes-14-00917-t003:** Longitudinal LMM examining the predictive values of the *KL*-VS^het^ and *APOE ε4* status on the yearly rate of cognitive decline in MMSE and CDR-SB scores in all participants (*N* = 665) and ADCOMS in ADNI participants (*n* = 178) with baseline age, sex, and years of education as covariates.

Cognitive Measure	Factor	*β* (*SE*)	*p* ^†^
MMSE	Time VS^het−^/*ε4−* v. VS^het−^/*ε4+* VS^het+^/*ε4−* v. VS^het−^/*ε4−* VS^het+^/*ε4+* v. VS^het−^/*ε4−* VS^het+^/*ε4−* v. VS^het−^/*ε4+ * VS^het+^/*ε4+* v. VS^het−^/*ε4+* VS^het+^/*ε4+* v. VS^het+^/*ε4−* VS^het−^/*ε4−* v. VS^het−^/*ε4+* x Time VS^het+^/*ε4−* v. VS^het−^/*ε4−* x Time VS^het+^/*ε4+* v. VS^het−^/*ε4−* x Time VS^het+^/*ε4−* v. VS^het−^/*ε4+* x Time VS^het+^/*ε4+* v. VS^het−^/*ε4+* x Time VS^het+^/*ε4+* v. VS^het+^/*ε4−* x Time	−1.535 (0.007) +0.169 (0.276) +0.027 (0.429) +0.027 (0.393) +0.197 (0.415) +0.196 (0.368) −0.001 (0.501) +0.290 (0.055) +0.287 (0.089) −0.325 (0.085) +0.577 (0.000) +0.036 (0.080) −0.613 (0.106)	0.000 0.540 0.949 0.945 0.635 0.594 0.999 0.000 0.001 0.000 0.000 0.659 0.000
CDR-SB	Time VS^het−^/*ε4−* v. VS^het−^/*ε4+* VS^het+^/*ε4−* v. VS^het−^/*ε4−* VS^het+^/*ε4+* v. VS^het−^/*ε4−* VS^het+^/*ε4+* v. VS^het−^/*ε4−* VS^het+^/*ε4+* v. VS^het−^/*ε4+* VS^het+^/*ε4+* v. VS^het+^/*ε4−* VS^het−^/*ε4−* v. VS^het−^/*ε4+* x Time VS^het+^/*ε4−* v. VS^het−^/*ε4−* x Time VS^het+^/*ε4+* v. VS^het−^/*ε4−* x Time VS^het+^/*ε4−* v. VS^het−^/*ε4+* x Time VS^het+^/*ε4+* v. VS^het−^/*ε4+* x Time VS^het+^/*ε4+* v. VS^het+^/*ε4−* x Time	+1.352 (0.040) −0.013 (0.187) −0.131 (0.286) +0.282 (0.267) −0.144 (0.276) +0.269 (0.251) +0.413 (0.336) −0.167 (0.031) −0.104 (0.046) +0.208 (0.047) −0.271 (0.045) +0.041 (0.045) +0.312 (0.057)	0.000 0.944 0.648 0.290 0.603 0.283 0.220 0.000 0.026 0.000 0.000 0.362 0.000
ADCOMS	Time VS^het−^/*ε4−* v. VS^het−^/*ε4+* VS^het+^/*ε4−* v. VS^het−^/*ε4−* VS^het+^/*ε4+* v. VS^het−^/*ε4−* VS^het+^/*ε4+* v. VS^het−^/*ε4−* VS^het+^/*ε4+* v. VS^het−^/*ε4+* VS^het+^/*ε4+* v. VS^het+^/*ε4−* VS^het−^/*ε4−* v. VS^het−^/*ε4+* x Time VS^het+^/*ε4−* v. VS^het−^/*ε4−* x Time VS^het+^/*ε4+* v. VS^het−^/*ε4−* x Time VS^het+^/*ε4−* v. VS^het−^/*ε4+* x Time VS^het+^/*ε4+* v. VS^het−^/*ε4+* x Time VS^het+^/*ε4+* v. VS^het+^/*ε4−* x Time	+0.128 (0.007) −0.008 (0.032) −0.137 (0.060) +0.020 (0.046) −0.145 (0.058) +0.029 (0.042) +0.117 (0.066) −0.033 (0.006) −0.042 (0.009) +0.024 (0.009) −0.075 (0.008) +0.009 (0.008) +0.066 (0.010)	0.000 0.791 0.023 0.662 0.012 0.500 0.080 0.000 0.000 0.005 0.000 0.243 0.000

(*SE*) = standard error. ^†^
*p* = 0.000 means *p* < 0.0005.

**Table 4 genes-14-00917-t004:** Averaged annualized rates of cognitive decline along with standard error given in parenthesis in participants grouped via cross-classification by *KL-*VS^het^ and *APOE ε4* status. Rates of yearly cognitive decline are derived from a multiple linear regression model.

	MMSE	CDR-SB	ADCOMS
VS^het−^/*ε4−*	−1.067 (0.049)	+1.056 (0.028)	+0.082 (0.005)
VS^het−^/*ε4+*	−1.423 (0.038)	+1.216 (0.022)	+0.112 (0.004)
VS^het+^/*ε4−*	−0.735 (0.081)	+0.903 (0.041)	+0.047 (0.004)
VS^het+^/*ε4+*	−1.358 (0.087)	+1.312 (0.046)	+0.122 (0.008)

**Table 5 genes-14-00917-t005:** Longitudinal LMM analyses investigating the effect of *KL-*VS^het+^ status for annualized rates of cognitive decline in *ε4* non-carriers stratified by sex, median baseline age, and median education level. For all comparisons, the reference group was *KL-*VS^het−^/*ε4−*.

			MMSE		CDR-SB	
Stratification	n		*β* (*SE*)	*p*	*β* (*SE*)	*p*
Female	310	VS^het+^/*ε4−*	+0.039 (0.152)	0.795	+0.015 (0.070)	0.834
Male	355	VS^het+^/*ε4−*	+0.368 (0.109)	0.001	−0.196 (0.061)	0.001 **
Baseline age ≤ 76 yr.	350	VS^het+^/*ε4−*	−0.029 (0.175)	0.867	+0.047 (0.095)	0.621
Baseline age > 76 yr.	315	VS^het+^/*ε4−*	+0.520 (0.101)	0.000	−0.285 (0.055)	0.000 **
Education < 16 years	231	VS^het+^/*ε4−*	−0.025 (0.177)	0.885	+0.136 (0.087)	0.117
Education ≥ 16 years	434	VS^het+^/*ε4−*	+0.287 (0.089)	0.001	−0.104 (0.046)	0.026

** indicates significance (*p* < 0.01) after Bonferroni correction for multiple testing.

## Data Availability

Demographic, cognitive, clinical, and genetic datasets of ADNI participants, which were analyzed in this study, are accessible from the Laboratory of Neuro Imaging (LONI) website at https://adni.loni.usc.edu (accessed on 23 January 2022). Demographic, cognitive, and clinical datasets of NACC participants, which were analyzed in this study, are accessible from the NACC website at https://naccdata.org (accessed on 5 July 2022), while corresponding genetic data are accessible from the website of the National Institute on Aging Genetics of Alzheimer’s Disease Data Storage Site at https://dss.niagads.org (accessed on 27 January 2022).

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
