# Peer review of "Interaction between KLOTHO-VS Heterozygosity and APOE ε4 Allele Predicts Rate of Cognitive Decline in Late-Onset Alzheimer’s Disease"

_genes, 2023, doi:10.3390/genes14040917_

Round 1
Reviewer 1 Report
Overall, this study is interesting. Considering the complexity of the neurobiology of Alzheimer’s disease, the heterogeneous patient population, and the limitations of currently available biomarkers, there is a need for more studies to address these limitations. In this regard, the findings of the current study are significant and will open possibilities for developing new biomarkers for AD.
However, there are a lot of errors in the manuscript and some areas are not clear. Addressing these shortcomings is very important before consideration for publication. Below are some of those concerns and suggestions.
There are two materials and methods section (1&2). Why? It’s supposed to be an introduction, I guess.
Figure 1 legend is there but couldn’t find any figure in the paper. There are no statements that describes about the figure1 in the results.
In materials and methods section 2.1 please include more information about the NACC and ADNI databases.
In section 2.3 needs more information and please explain more about the measures MMSE, CDR-SB, ADAS in the methods section or in the results section. Please explain about the purpose of these tests and the describe how it is performed.
Section 3.2 please explain which sex used for those analysis if its mixed cohorts please explain detailly.
Section 3.3 The protective effect of KL-VS heterozygosity is observed in males, in individuals with an older age of cognitive decline onset, and in those with 16 or more years of education. Just curious to know why this outcome only from males and why its not similar if females? Any explanation would be appropriated.
Some abbreviations are not consistent, please check once again throughout the manuscript.
I recommend the authors to convert the tables to bar graphs with p value*s. that will easy to understand.
Also Table 1 there is no details about the significance. Significant between which groups or among the groups? Please explain and mention it in the table legends.
Please explain how the * p values has been determined. Table 1 (0.001) given one *, in the table 5 (0.01) given similar one *. Why one * given to different p values. Number of * should not be different?
Also, authors mentioned 0.000, please give values.
Author Response
Reviewer #1
- Overall, this study is interesting. Considering the complexity of the neurobiology of Alzheimer’s disease, the heterogeneous patient population, and the limitations of currently available biomarkers, there is a need for more studies to address these limitations. In this regard, the findings of the current study are significant and will open possibilities for developing new biomarkers for AD.
We greatly value the positive assessment of our work by the reviewer.
- However, there are a lot of errors in the manuscript and some areas are not clear. Addressing these shortcomings is very important before consideration for publication. Below are some of those concerns and suggestions. There are two materials and methods section (1&2). Why? It’s supposed to be an introduction, I guess.
We thank the reviewer for noting this error, the Introduction has been properly labeled.
- Figure 1 legend is there but couldn’t find any figure in the paper. There are no statements that describes about the Figure 1 in the results.
We apologize for the figure being inaccessible, but this most likely was an issue related to the manuscript submission system. To avoid similar problem with the R-1 resubmission we included jpeg versions of Figure 1 (and newly added Figure 2) at the end of the word manuscript file. To the address the second issue we have added the following statement in the Results section: “These interactions between KL-VShet+ and APOE ε4 allele are illustrated for the entire mixed-sex cohort in three linear regression plots divided by cognitive measure in Figure 1 A-C.”
- In materials and methods section 2.1 please include more information about the NACC and ADNI databases.
We acknowledge the reviewer’s request and have added the following statement in the Materials and Methods section to better describe the NACC database: “The UDS is a data repository that contains annual clinical evaluations of participants who were recruited by NIA Alzheimer's Disease Researcher Centers (ADRC) programs starting in 2005 and continuing to the present day. Each program has its own protocol for referrals and recruitment, and a more detailed summary of NACC data compilation is available at https://naccdata.org/requesting-data/data-request-process#naccHandbook.” As well, we added the following statement for the ADNI database: “ADNI-1 (2004-2009), ADNI-GO (2009-2011), ADNI-2 (2011-2016), and ADNI-3 (2016-present). These are successive studies that emphasize participant rollover with additional recruitment goals for each study. Complete ADNI criteria and data collection information are available at http://adni.loni.usc.edu/data-samples/clinical-data/ and https://adni.loni.usc.edu/methods/documents/.”
- In section 2.3 needs more information and please explain more about the measures MMSE, CDR-SB, ADAS in the methods section or in the results section. Please explain about the purpose of these tests and the describe how it is performed.
We appreciate the reviewer’s comment and in response we have added the following explanation for MMSE and CDR-SB measures: “The MMSE assesses cognitive function across several domains domains including orientation to time and place, registration, attention and calculation, short-term recall, visuospatial functions, and language skills to screen for cognitive impairment and to approximate its depth. The test is performed by a trained healthcare professional who asks a battery of questions and provides a series of tasks that take approximately 15 minutes to complete. The CDR serves a similar purpose; however, it evaluates cognition across six domains (memory, orientation, judgement and problem solving, community affairs, home and hobbies, and personal care) based on a semi-structured interview that includes responses from the participant and the participant’s caregiver. CDR-SB is then computed by summing the domain box scores”. As well as the following statement for the ADAS-11 measure: “…which evaluates cognition through a series of 11 tasks and questions including word recall, orientation, and language comprehension.”
- Section 3.2 please explain which sex used for those analysis if its mixed cohorts please explain detailly.
We thank the reviewer for bringing up the lack of clarity concerning the sex composition of the compared cohorts in section 3.2. For this initial linear mixed model analysis, we used a mixed-sex cohort detailed in Table 1. To address the reader’s point, we have added the following statement at the beginning of section 3.2: “The analysis using longitudinal linear mixed models was conducted on a cohort consisting of both male and female participants, with roughly equal numbers of each sex. This initial cohort was divided based on KL-VShet+ and ε4 carrier status, as detailed in Table 1.”
- Section 3.3 The protective effect of KL-VS heterozygosity is observed in males, in individuals with an older age of cognitive decline onset, and in those with 16 or more years of education. Just curious to know why this outcome only from males and why its not similar if females? Any explanation would be appropriated.
We recognize the reviewer’s question about the sex-stratified nature of the results, and have provided a series of possible explanations in the Discussion section, starting at the bottom of page 17.
- Some abbreviations are not consistent, please check once again throughout the manuscript.
We thank the reviewer for pointing out these unintentional omissions. In response to the reviewer’s comment, we have edited the manuscript for consistency in abbreviation use.
- I recommend the authors to convert the tables to bar graphs with p value*s. that will easy to understand.
We have deeply considered the reviewer’s comment and have determined that the information in the manuscript’s tables cannot be easily conveyed by bar graphs. Such a conversion would obscure overview of the detailed results and would not improve readability of the manuscript.
- Also Table 1 there is no details about the significance. Significant between which groups or among the groups? Please explain and mention it in the table legends.
We thank the reviewer for this note, and in the caption for Table 1 we have clarified, which groups the significance refers to.
- Please explain how the * p values has been determined. Table 1 (0.001) given one *, in the table 5 (0.01) given similar one *. Why one * given to different p values. Number of * should not be different? Also, authors mentioned 0.000, please give values.
We thank the reviewer for pointing out these inconsistences concerning labeling of p values across the Tables. These were revised and made consistent. Re. the value of “0.000” given in Table 3, it results from the limitations of SPSS software and its analysis outputs, which provides p-values with accuracy of three decimal places. In response to the reviewer query we included in the caption of Table 3 an information that the value of “0.000” reflects p values lower than 0.0005.
Reviewer 2 Report
I have read carefully the manuscript entitled “Interaction between KLOTHO-VS Heterozygosity and APOE ε4 allele Predicts Rate of Cognitive Decline in Late-Onset Alzheimer’s Disease”. The methodology is generally appropriate, well-presented and organized in a logical way. The authors concluded their work by stating that KL-VShet+ status has a protective effect on AD progression and interacts with the e4 allele . Some points should be addressed before the manuscript can be considered for publication.
On page 3 the paragraph was incorrectly called “Materials and methods”. The authors are asked to correct the title of the paragraph.
The article refers to the databases from which the authors drew the information for this article where all the important informations regarding the protocols of the data collections are present. However, to make it easier to read and more immediate for readers to understand, it would be better to specify more directly and more clearly some details in the text:
- The protocols of the data collections you refer to indicate through which tests the determination of cognitive decline was made, however, it would be better to mention them in the text of the article as well.
- The authors stated that the initial diagnosis was "mild cognitive disorder," and the later diagnosis was Alzheimer's disease. After how long did the diagnosis change? Furthermore, in the study the authors mention 3 follow-up, but do not specify after how long they were carried out. It would be interesting to include this data as well.
- In the protocols for the databases used, the inclusion and exclusion criteria are well specified, however, it would be better to specify also in the text of the article that all those patients whose past illnesses could alter the results of the study were excluded from enrollment.
To improve understanding of the study's conclusions, the authors should add two linear graphs, one for male participants and one for female participants. The purpose of these graphs should be to summarize the results of the data in the tables, highlighting how genotype correlates with slower or more rapid cognitive decline over time. For example, the graphs might report on the x-axis the years of follow up and on the y-axis the proportion developing dementia. Each line on the graph should represent a gene profile.
Author Response
- I have read carefully the manuscript entitled “Interaction between KLOTHO-VS Heterozygosity and APOE ε4 allele Predicts Rate of Cognitive Decline in Late-Onset Alzheimer’s Disease”. The methodology is generally appropriate, well-presented and organized in a logical way. The authors concluded their work by stating that KL-VShet+ status has a protective effect on AD progression and interacts with the e4 allele. Some points should be addressed before the manuscript can be considered for publication.
On page 3 the paragraph was incorrectly called “Materials and methods”. The authors are asked to correct the title of the paragraph.
We deeply appreciate the reviewer’s affirmative judgment, and have addressed the issue about titling the introduction.
- The article refers to the databases from which the authors drew the information for this article where all the important information regarding the protocols of the data collections are present. However, to make it easier to read and more immediate for readers to understand, it would be better to specify more directly and more clearly some details in the text:
- The protocols of the data collections you refer to indicate through which tests the determination of cognitive decline was made, however, it would be better to mention them in the text of the article as well.
We thank the reviewer for this suggestion, and in response we have added the following statement to section 2.1 detailing which assessments were used in both the ADNI and the NACC as the basis for MCI and AD dementia diagnosis: “In the ADNI, subjects were diagnosed with MCI if there was a memory complaint by either the participant or the study partner, memory loss measured by education-adjusted scores on the Weschler Memory Scale (Logical Memory II subscale), a score between 24 and 30 (inclusive) on the MMSE, a score of 0.5 on the CDR, and preserved general cognition and functional performance, as determined by the site physician. A diagnosis of AD was recorded if participants scored between 20 and 26 (inclusive) on the MMSE, either 0.5 of 1.0 on the CDR, and met the NINCDS/ADRDA criteria for probable AD. In the NACC, clinicians were instructed to assess cognition with neuropsychological testing of their choosing, only providing commonly used cut-off points such as a score of 0.5 on the CDR representing MCI and a score of 1.0 or above representing AD.”
- The authors stated that the initial diagnosis was "mild cognitive disorder," and the later diagnosis was Alzheimer's disease. After how long did the diagnosis change? Furthermore, in the study the authors mention 3 follow-up, but do not specify after how long they were carried out. It would be interesting to include this data as well.
We appreciate the reviewer’s question, which has been answered in section 3.1 where we state that the mean baseline age (i.e. entry to the study with the initial diagnosis of MCI) is 75.5 years, while the transition age (i.e. first time diagnosis of AD dementia) is 77.7 years. Thus, in the interrogated cohort, the average time to transition from MCI to AD dementia (the diagnosis change) is approximately 2.2 years. The total length of follow up across the entire study is closely reflected by the number of visits, which is on average 10.7 ± 4.1 (± standard deviation) (Table 1). These visits always occurred at yearly interval, with the exception of the second ADNI visit, which took place at 6 months after the baseline visit. The “3 follow-up” refers to the minimum number of follow up visits during the AD dementia stage, which was set up as a prerequisite for study inclusion. Thus, we included only individuals having at least three follow-up visits after receiving an AD dementia diagnosis, which means they were followed with this diagnosis for at least 2 years. Given that the average total study follow up was longer than 10 years, the number of individuals who exactly met but did not exceeded this criterion was minute.
- In the protocols for the databases used, the inclusion and exclusion criteria are well specified, however, it would be better to specify also in the text of the article that all those patients whose past illnesses could alter the results of the study were excluded from enrollment.
The reviewer raises here a valid point which we addressed in the section 2.1 of the revised manuscript “Both the ADNI and NACC also consider other neurological conditions that may contribute to or directly cause MCI or dementia. Participants with these neurological co-morbidities were excluded from analysis”. Given the average follow up time across the entire study exceeded 10 years, enrolled participants were generally healthy, and their chronic non-neurological conditions e.g. hypertension or hyperlipidemia (if present) did not interfere with study participation.
- To improve understanding of the study's conclusions, the authors should add two linear graphs, one for male participants and one for female participants. The purpose of these graphs should be to summarize the results of the data in the tables, highlighting how genotype correlates with slower or more rapid cognitive decline over time. For example, the graphs might report on the x-axis the years of follow up and on the y-axis the proportion developing dementia. Each line on the graph should represent a gene profile.
We thank the reviewer for the excellent suggestion, and have incorporated four additional graphs in Figure 2 that include the sex-stratified scatterplots for MMSE and CDR-SB, which clearly visualize the impact of sex on the protective effect of KL-VShet+ status in the APOE ε4 non-carriers.
Round 2
Reviewer 2 Report
The requested revisions have all been carried out correctly. I believe the manuscript is ready for publication